# The Senotherapeutic Effects of APPA (Apocynin [AP] and Paeonol [PA]) on Senescent Human Chondrocytes

**DOI:** 10.3390/ph18091386

**Published:** 2025-09-16

**Authors:** Mercedes Fernández-Moreno, Tamara Hermida-Gómez, Carlos Vaamonde-Garcia, Sara Paniagua-Barro, Nicholas Larkins, Alan Reynolds, Francisco J. Blanco

**Affiliations:** 1Grupo de Investigación en Reumatología (GIR), Instituto de Investigación Biomédica de A Coruña (INIBIC), Complexo Hospitalario Universitario de A Coruña (CHUAC), Sergas, Universidade de A Coruña (UDC), 15005 A Coruña, Spain; tamara.hermida.gomez@sergas.es (T.H.-G.); carlos.vaamonde.garcia@sergas.es (C.V.-G.); sara.paniagua.barro@sergas.es (S.P.-B.); 2Grupo de Investigación en Reumatología y Salud (GIR-S), Centro Interdisciplinar de Química y Biología (CICA), Universidade de A Coruña (UDC), Campus de Elviña, 15071 A Coruña, Spain; 3Centro de Investigación Biomédica en Red, Bioingenieria, Biomaterial y Nanomedicina (CIBER-BBN), 28029 Madrid, Spain; 4AKL Therapeutics Ltd., Stevenage Bioscience, Gunnels Wood Rd, Stevenage SG1 2FX, UK; nl@aklrd.com (N.L.); ar@aklrd.com (A.R.); 5Grupo de Investigación en Reumatología y Salud (GIR-S), Departamento de Fisioterapia, Medicina y Ciencias Biomédicas, Facultad de Fisioterapia, Centro Interdisciplinar de Química y Biología (CICA), INIBIC-Sergas, Universidade de A Coruña (UDC), Campus de Oza, 15006 A Coruña, Spain

**Keywords:** chondrocyte, APPA, senescence, apoptosis, proliferation, senotherapeutic

## Abstract

**Background/Objectives**: Osteoarthritis (OA) is a complex joint disease involving chronic inflammation, aging, and obesity, affecting nearly 6 million people worldwide. Senescent cells in OA are linked to increased inflammation, oxidative stress, and DNA damage, making them potential therapeutic targets. APPA, a combination of apocynin (AP) and paeonol (PA), has shown anti-inflammatory and antioxidant properties. This study evaluated the effects of APPA on cellular senescence in human articular chondrocytes. **Methods**: Using a chondrocyte cell line (T/C-28a2) and primary human chondrocytes, senescence was induced with etoposide and Oncostatin M (Eto + OSM), followed by treatment with APPA, AP, or PA. Senescence markers (SA-β-gal, *P21*_*CDKN1A*_), apoptosis, proliferation (Ki67), and rps6 protein levels were analyzed. **Results**: APPA significantly reduced SA-β-gal activity and *p21* expression in cell model—effects not replicated by AP or PA alone. APPA increased early apoptosis and dual-labeled senescent-apoptotic cells, along with total cell numbers and rps6 levels. It also altered Ki67 expression in different cell subpopulations, suggesting effects on proliferation. **Conclusions**: This study suggests that APPA exerts senotherapeutic effects on human senescent chondrocytes. A reduction in SA-β-gal together with an increase in cell numbers and the proliferation marker Ki67 suggests possible senomorphic effects, whereas a reduction in SA-β-Gal accompanied by an increase in apoptosis indicates senolytic activity. These findings support recent evidence that the distinction between senolytic and senomorphic agents is ‘fuzzy’.

## 1. Introduction

Osteoarthritis (OA) is the most common musculoskeletal condition, affecting over 595 million people worldwide [1]. Historically, OA has been considered to result from ‘wear and tear,’ leading to cartilage loss and reductions in the joint space, resulting in pain (the dominant symptom of OA) with a substantial socioeconomic burden [2]. Recent research has established that OA involves the whole joint with not only the loss of cartilage, but also changes in the subchondral bone, synovium, tendons, ligaments, and muscles as a result of chronic inflammation [3], mainly involving the innate immune system [4]. This inflammation may be triggered, for example, by aging or obesity [5] and has been termed ‘inflammaging’ [6], which has been linked to cell senescence and, amongst other diseases, OA [7]. It is now accepted that OA is a heterogenous disease with several different phenotypes, endotypes, and possible triggers that lead to a common final pathway resulting in joint destruction [8,9,10,11,12,13]. The roles of bone, cartilage, and synovium in OA, with subsequent crosstalk between them involving many different pathways, provide numerous potential treatment targets [14,15,16]. This understanding has led to the proposal that effective treatments will need a multifaceted approach [17].

Cellular senescence is a stress-induced response that results in permanent cell cycle arrest and significant phenotypic alterations, including the production of a bioactive secretome known as the senescence-associated secretory phenotype (SASP) [18]. Both intracellular and extracellular signals can drive cells into this senescent state, predominantly those associated with tissue or cellular damage [19,20]. The presence and accumulation of senescent cells in tissues contributes to age-related diseases [21,22,23], and chondrocyte senescence is thought to play a role in the initiation and progression of OA [24]. Senescent cells from OA patients showed increased expression of nitric oxide, pro-inflammatory cytokines p21 (*Cyclin Dependent Kinase Inhibitor 1A* (*CDKN1A*)) and p16 (*Cyclin Dependent Kinase Inhibitor 2A* (*CDKN2A*); activated DNA damage response; reactive oxygen species (ROS) production; and senescence associated β-galactosidase (SA-β-Gal) activity [21,25].

Consequently, targeting senescent cells to delay aging and the adverse effects of cellular senescence is becoming an increasingly popular area of research [26,27,28]. During our earlier study, published in 2024 [29], we had found that APPA reduced the numbers of senecent cells [30], which led us to undertake the program of work described here.

Senotherapeutics are a class of drugs that specifically target and modulate senescent cells. They consist of senomorphics that suppress either all or several senescent cell characteristics by blocking SASP without inducing cell death and senolytics, which selectively kill senescent cells through apoptosis. Advances over the last 30 years have laid the groundwork for novel therapies, including senolytics and senomorphics, which show potential for modifying OA progression [23]. Senotherapeutic agents being evaluated for treatment of OA include curcumin, dasatinib, quercetin, fisetin, navitoclax, and UBX0101; so far, none have proved effective in clinical trials [31,32].

OA treatment guidelines recommend non-pharmacological strategies as the first line of therapy, including exercise, patient education, and weight loss [33,34,35]. Pharmacological treatment options are limited and only provide symptom relief that, when this becomes intolerable, leads to surgical interventions such as total joint replacement. Currently there are no treatments capable of altering the natural progression of OA. These potential treatments, referred to as disease-modifying osteoarthritis drugs (DMOADs), remain under investigation, with several candidates currently in various phases of clinical development [36,37,38].

APPA is a synergistic combination of two bioactive phenolic isomers, apocynin (4-hydroxy-3-methoxyacetophenone) (AP) and paeonol (2′-hydroxy-4′-methoxyacetophenone) (PA), developed by a series of n = 1 studies in dogs with naturally occurring OA, undertaken by one of the authors (NL). Both these compounds are derived from traditional medicinal plants and exhibit a wide range of overlapping and complimentary pharmacological effects [39,40,41]. In combination as APPA they have been found to scavenge ROS and reduce or inhibit the release of several inflammatory cytokines and matrix metalloproteinases that are known to be components of the SASP; APPA also protects against proteoglycan loss from cartilage explants [42,43,44,45]. Both AP and PA have shown efficacy in OA using in vitro and animal models [46,47,48,49]. In the rat meniscus tear model of OA, APPA has also been shown to be effective [50,51] with evidence of synergy (unpublished). Randomized controlled trials in dogs with naturally occurring OA have provided evidence for the efficacy of APPA in this disease [52,53,54]. In a recent Phase 2a study, oral administration of APPA demonstrated both efficacy and safety, significantly reducing pain compared to placebo in a subset of patients with knee osteoarthritis [55].

**The aim of this study** was to evaluate the effects of APPA, AP, and PA on senescent human chondrocytes and determine whether these effects were senotherapeutic.

## 2. Results

### 2.1. Evaluation of Stimuli to Induce Senescence

Data showed that when T/C-28a2 cells were incubated with either interleukin-6 (IL-6, 20 ng/mL) or etoposide (Eto, 5 µM), both in the presence of oncostatin M (OSM, 10 ng/mL), the number of senescent cells increased, but only in the case of Eto + OSM was this significant (Figure 1A). Therefore 5 µM Eto combined with 10 ng/mL OSM (Eto + OSM) during 48 h was chosen for subsequent experiments with T/C-28a2 cell line.

When senescent T/C-28a cells induced by Eto + OSM were treated with APPA for 24 h, SA-β-gal activity was significantly reduced in comparison to that in cells incubated with Eto + OSM alone (Figure 1B).

To confirm this result, we evaluated effects of APPA using human primary chondrocytes. The optimal concentration of Eto was determined in human primary chondrocytes and found to be a higher concentration than for the cell line, 20 µM in combination with 10 ng/mL OSM for 48 h, which significantly increased both senescence markers evaluated, SA-β-gal activity and p21 (*CDKN1A*) gene expression (Figure 1C). The analysis of senescence in primary human chondrocytes demonstrated that APPA decreases senescent cells with SA-β-gal activity and p21 (*CDKN1A*) gene expression in cells incubated in the presence of Eto + OSM (Figure 1D).

### 2.2. Effect of APPA and Its Components on Senescent T/C28a2 Chondrocytes

Having shown that APPA decreased senescent markers, we then investigated the effects of each component of APPA, AP, and PA, independently.

In our previous paper in this journal, we had established that 10 µg/mL APPA did not affect chondrocyte viability [29], and that APPA is a combination of apocynin (AP) and paeonol (PA) in a 2:7 ratio. The effect of individual concentrations of AP (2.3 and 10 µg/mL) and PA (7.7 and 10 µg/mL) on T/C28a2 chondrocyte viability were tested using the DRAQ7 assay. A dot plot from the DRAQ7 assay was used to estimate cell viability in the presence of both drug concentrations during 24 h of incubation. Analysis of these data indicated that the individual components, AP (2.3 and 10 µg/mL) and PA (7.7 and 10 µg/mL), did not significantly affect cell viability (Figure 2A). The percentage of non-viable cells never exceeded 7.05%. Based on these findings, subsequent experiments were designed using APPA, AP, and PA at the concentrations described.

To evaluate whether APPA or its individual components independently influenced cellular senescence, SA-β-gal activity was analyzed. The results showed that APPA reduced the proportion SA-β-gal positive cells induced by Eto + OSM, whereas this effect was not observed with either concentration of AP or PA (Figure 2B). These findings indicate that both components of APPA are required to significantly decrease SA-β-gal activity in cultures exposed to Eto + OSM.

When the senescence-modulating activity of APPA and its individual components was evaluated in human primary chondrocytes cultured in presence of Eto + OSM, the data revealed that SA-β-gal activity and *CDKN1A* gene expression were reduced in the presence of APPA. However, no modulation was observed when either AP or PA was evaluated individually (Figure 2C,D). These findings further confirm that both components of APPA are required to reduce the senescence markers where Eto + OSM was used to induce senescence.

### 2.3. APPA Increased the Number of T/C28a2 Apoptotic Cells

The analysis of T/C28a2 apoptotic cells showed that Eto + OSM, APPA, AP, and PA did not modulate this pathway in comparison to untreated cells (Appendix A). When senescence was induced, data showed that all concentrations of APPA, AP, and PA significantly increased the percentage of cells in early apoptosis (Figure 3). These results indicate that APPA, AP, and PA exhibited the ability to increase apoptotic cells under senescence-inducing conditions. Autophagy was also evaluated by WB analysis; however, data obtained demonstrated that these compounds did not modulate this process (Appendix A).

Given that the data showed that APPA reduces the proportion of senescent cells while increasing apoptotic cells in senescence-induced conditions, we conducted an analysis of cells with dual markers for senescence and apoptosis. This approach allowed us to determine the percentage of cells simultaneously expressing both senescence and apoptotic markers (Figure 4). The effects of APPA, AP, and PA on the cells without treatment with Eto + OSM were evaluated, and the data revealed that none of the compounds tested had any effect under these conditions (Appendix A). The data showed that Eto + OSM increased the percentage of cells positive for both FDG and Annexin-V double markers compared to the basal condition (Figure 4A).

The evaluation of positive cells for FDG and Annexin-V in a senescence environment treated with APPA, AP, and PA was carried out. The analysis showed a significant increase in double-marker-positive cells in senescent cells treated with APPA, AP, or PA compared to the Eto + OSM condition (Figure 4B). These findings highlight the ability of APPA to enhance the apoptotic process in senescent cells. Altogether, the data demonstrate that APPA exerts a senotherapeutic effect, as this combination reduced the number of senescent cells by inducing apoptosis.

### 2.4. Evaluation APPA Effect on Cell Proliferation

T/C28a2 cells incubated in the presence of Eto + OSM and APPA showed an increased number of cells compared to those treated with Eto + OSM alone, as indicated by Hoechst 33342 staining (Figure 5A). These results were confirmed by the higher levels of p-rps6 in cells incubated with Eto + OSM compared to the basal condition. Furthermore, in comparison to the cells treated with only Eto + OSM, those treated with both Eto + OSM and APPA showed an increase in p-rps6 protein levels, while the effects of AP and PA were not significant (Figure 5B). APPA, AP, and PA, used on the cells without Eto + OSM treatment, showed lower p-rps6 modulation (Appendix A).

The percentage of proliferating cells was further investigated by detecting the number of Ki67-positive cells. Two distinct populations (P1 and P2) were identified in the dot-plots, which were analyzed independently (Figure 5C).

Analysis of population P1 revealed that the percentage of Ki67-positive cells decreased in the presence of Eto + OSM compared to the basal condition. Furthermore, when comparing cells treated with Eto + OSM alone to those treated with both Eto + OSM and APPA, this reduction was even more pronounced; however, AP and PA had no effect. Cells in population P2 exhibited the opposite behavior. The percentage of cells in this population was very low under basal conditions; however, the addition of Eto + OSM significantly increased the number of cells. Moreover, the presence of APPA, AP, and PA in Eto + OSM-treated cells further increased the percentage of Ki67-positive cells in a statistically significant manner (Figure 5C).

In summary of these data, Ki67 distinguished two populations of T/C-28a2 chondrocytes (P1 and P2), which showed clearly different behaviors under the experimental conditions. In population P1, the percentage of Ki67-positive cells decreased in the presence of Eto + OSM compared with basal conditions, and this reduction was even more pronounced with Eto + OSM + APPA. In contrast, population P2—which was almost absent under basal conditions but increased after Eto + OSM treatment—showed a significant rise in Ki67-positive cells when APPA was added, consistent with enhanced proliferative activity. These results suggest that APPA may have the capacity to enhance the proliferative potential of non-senescent cells. However, further studies are required to confirm APPA’s effect on non-senescent cells in a senescence inducing environment.

## 3. Discussion

APPA is a fixed combination of apocynin (AP) and paeonol (PA), both of which have been shown to have a wide range of pharmacological effects, including ROS scavenging and inhibition of inflammation, which have been demonstrated in cells, tissues, and animal models of disease [39,40,41,44]. APPA holds promise in several other therapeutic areas due to the combined effects of AP and PA, having pharmacological applications in cardiovascular, inflammatory, neurodegenerative diseases, OA, and cancer [29].

In a recent clinical trial in patients with knee OA, APPA was found to be safe and well-tolerated, and although no significant symptom improvement in pain was seen in the overall population, significant benefits were seen in a pre-determined subgroup of patients with nociplastic pain and more severe OA [55]. In the rat meniscal tear model of OA, APPA has shown evidence of disease modification [51] and provided symptom relief and functional improvements in naturally occurring canine OA [52,54].

In our previous publication in this journal, APPA was shown to impact pathways known to be involved in the pathogenesis of OA; in human articular chondrocytes stimulated with IL-1β, APPA decreased oxidative stress and reduced the expression of TNF-α, IL-8, MMP-3, and MMP13, all markers of SASP. These effects translated into less proteoglycan loss in the intermedial layer of explants and lower levels of glycosaminoglycan release into the supernatant of the explants [29]. In human neutrophils, APPA has also been shown to reduce inflammatory pathways, probably through inhibition of NF-κB activation, resulting in downregulation of inflammatory cytokines, including IL-1β, IL-6, IL-8, and TNF-α; reductions in oxidative stress and reactive oxygen species (ROS) were related to the upregulation of Nrf2 [56].

Although various cell types are involved in OA pathology, chondrocytes are primarily thought to play a major role in OA induction through cellular senescence [24]. Chondrocyte senescence is believed to be closely linked to OA, which also progresses with age. Additionally, OA is associated with chondrocyte proliferation, telomere shortening, and SA-β-gal expression [57]. Two different mechanisms of senescence are suggested in chondrocytes: (1) replicative senescence and (2) stress-induced premature senescence (SIPS) [58].

In our model, chondrocytes were treated with Eto + OSM to induce cellular senescence. The senescence-inducing effect of Eto + OSM in cell cultures, beyond inhibiting cell growth, is evidenced by the presence of several markers of cellular senescence, such as enlarged nuclei, activated SA-β-gal activity, elevated levels of p53 and *CDKN1A*, down-regulation of Lamin B1, or the presence of multiple centrosomes, which have been observed in several studies [59,60]. In this study, some of these parameters were evaluated, and chondrocytes incubated in the presence of Eto + OSM exhibited higher SA-β-gala activity and increased *CDKN1A* levels compared to untreated chondrocytes. Approximately 20% of cells showed the presence of senescent markers. However, etoposide exhibits multiple effects on cells beyond senescence induction. This compound, a genotoxic agent widely employed in cancer treatment, induces DNA double-strand breaks through the inhibition of topoisomerase II. In addition, it also modulates immune responses by regulating the expression of different cytokines and immune mediators. The combination of etoposide with adjunctive agents has shown significant efficacy in mitigating cytokine storms in patients with COVID-19 and other conditions [61,62]. Due to this effect, it has also been used in patients with lupus and secondary rheumatoid arthritis [63]. However, the molecular mechanisms driving these immunomodulatory effects remain poorly understood [64,65].

Defining senescent cells, particularly in vivo, is challenging due to the lack of specificity of individual markers like SA-β-gal, which is especially problematic in cultured cells [66,67]. A more robust approach involves analyzing multiple senescence markers to identify subpopulations enriched with aging and reduced by senescent cell clearance [68]. In OA samples, chondrocytes expressing senescence markers appear heterogeneous [69]. A recent study using single-cell analytical techniques has revealed that there are diverse chondrocyte subtypes with distinct roles in cartilage health and disease [70]. We acknowledge that additional complementary assays, including p16/protein-level analysis, cell cycle profiling, and SASP biomarker evaluation and single-cell techniques, would further strengthen confirmation of stable senescence.

APPA reduced senescent cells induced in chondrocytes incubated with Eto + OSM, as evidenced by β-gal activity and *p21* gene expression analysis. Additionally, our results showed that APPA increased the number of apoptotic cells in a senescent environment. However, an additional challenge in identifying senescent cells stems from their dynamic nature, as the expression of specific markers above baseline levels can vary depending on the stage or degree of senescence [23]. These findings open a future line of research that should be considered in upcoming studies, as it is crucial, based on current research, to focus efforts on determining the degree of senescence in the cells used in in vitro models.

Our data suggest that APPA exerts this beneficial effect when used as a combined formulation, as its individual components (AP and PA) did not produce the same response in senescent cells. The combination of AP and PA in APPA has been shown to provide broader and more potent modulation of inflammatory and immunoregulatory pathways than either compound alone [29,55]. These data highlight the necessity of using APPA to achieve its effects on cellular senescence, rather than using its individual components separately, as AP and PA did not show a significant effect in reducing the analyzed senescence markers, p-rpS6 levels, or Ki67.

Senolytic and senomorphic therapies have garnered increasing attention over the past decade. These therapies are designed to selectively target and eliminate senescent cells without affecting healthy, ‘normal’ cells [71]. When we analyzed the mechanism of senescent cell elimination using APPA, our data showed an increased percentage of cells positive for both the senescence marker and Annexin-V in the presence of Eto + OSM. These findings suggest that APPA reduces senescent cell burden through apoptosis. The data obtained indicated that APPA exhibited this senolytic effect on senescent chondrocytes.

Discrimination between quiescence and senescence was based on the expression of Ki-67, a protein that is associated with cellular proliferation and a ribosomal protein S6 (rps6), a marker of active protein synthesis that is present in senescent cells but absent in quiescent cells. The expressions of Ki67 and rps6, along with the evaluation of β-gal activity, are the most widely used “tools” to identify quiescent, senescent, and stressed cells [72]. Our data revealed that chondrocytes incubated with Eto + OSM and APPA resulted in an increase in the number of cells, as well as elevated levels of p-rps6 and decreased Ki67 in population P1; however, in population P2, cells showed an increase in Ki67. In the literature, the discrimination between quiescence and senescence is based on the expression of ribosomal protein S6 (rps6), a marker of active protein synthesis that is present in senescent cells but absent in quiescent cells. Focusing on this data, we can confirm that at least a proportion of our chondrocytes are senescent. Upon activation of protein synthesis by extracellular or intrinsic stimuli, rps6 is phosphorylated (p-rps6) by S6 kinases that belong to the mTOR signaling pathway. P-rps6 is considered an indicator of active protein synthesis and has been proposed as a marker that allows discrimination between senescent and quiescent cells [73].

Population P1, as defined in this article, showed SA-β-gala activity (+), p-rps6 (+), and Ki67 (−), while Population P2 showed SA-β-gala activity (−), p-rps6 (+/−), and Ki67 (+), reflecting two different states: P1 could be senescent cells and P2 cycling stress or even pre-senescent cells [72]. Based on our data, we estimate that APPA enhances the proliferative capacity in a subset of chondrocytes incubated with Eto + OSM. Since not all cells were in a fully senescent state, they were likely in a state of cellular stress or pre-senescence.

Literature data indicate that the activation of growth-promoting pathways, alongside cell cycle arrest, drives cellular senescence, whereas quiescence could inhibit this process [73,74]. Growth stimulation, particularly via the mTOR pathway, can induce cellular senescence when accompanied by cell cycle arrest. In this context, senescence arises because growth-promoting signals, even in the absence of cell division, drive cellular hypertrophy and other senescence-associated phenotypes. Notably, while cell cycle inhibitors such as CDKN1A (p21) can halt proliferation, they do not suppress mTOR activity, allowing continued growth signaling that contributes to the senescent state [74].

These data indicate that at least a fraction of chondrocytes exposed to Eto + OSM entered senescence and responded to APPA by enhancing apoptotic pathways, consistent with senolytic activity. In contrast, another subset of chondrocytes, likely in a different cellular state, responded by increasing Ki67 expression, suggesting a proliferative recovery, more aligned with senomorphic effects. Taken together, our results suggest that APPA exerts senotherapeutic activity in senescent human chondrocytes through a combination of senomorphic and senolytic mechanisms. The reduction in SA-β-Gal activity together with increased cell numbers and Ki67 expression implies a senomorphic action, whereas the decrease in SA-β-Gal-positive cells accompanied by enhanced apoptosis, particularly under Eto-OSM-induced senescence, reflects a potential senolytic effect. These dual responses align with recent perspectives that the boundary between senomorphic and senolytic activities is diffuse or “fuzzy” [75] and therefore support the classification of APPA as a potential senotherapeutic agent with dual activity.

## 4. Material and Methods

### 4.1. Drug Preparation

APPA was supplied by AKL Research and Development Ltd. (Manchester, UK) in a prepared vial (2:7 ratio of AP:PA), while AP and PA were obtained from Sigma (Sigma-Aldrich, St Louis, MO, USA). All compounds were dissolved in Dimethyl Sulfoxide (DMSO) (Sigma-Aldrich, St Louis, MO, USA) at a final concentration of 1 g/mL, from which serial dilutions were developed. The 2:7 AP:PA ratio was established in preclinical studies as the most effective proportion and subsequently validated in a Phase 2a clinical trial in patients with knee osteoarthritis [51,52,55]. Animal studies and n = 1 studies in dogs with naturally occurring OA also demonstrated that this ratio produced optimal therapeutic effects, guiding its selection for clinical testing.

### 4.2. Chondrocytes Culture

#### 4.2.1. Chondrocytes from Human OA Hip Cartilage

Chondrocytes from human OA hip cartilage were isolated as previously described [76] from 18 patients with total hip arthroplasty (OA grade IV) (5 male with a mean ± SD age of 86 ± 7.17 years and 13 female with a mean ± SD age of 79.71 ± 13.56 years). Samples were obtained from the Sample Collection for Research on Rheumatic Disease, started by Dr. Francisco Blanco-García. This collection was authorized by the Galician Research Ethics Committee with registry code 2013/107 and has been registered with the National Registry of Biobanks, Collections Section code: C.0000424. Written informed consent was obtained from all the subjects involved in the study. All the procedures were conducted according to the principles expressed in the Helsinki Declaration of 1975, as revised in 2000.

The chondrocytes were isolated after surgery. Briefly, cartilage sections were aseptically removed from each donor and enzymatically digested with trypsin (Sigma-Aldrich, St Louis, MO, USA) for 15 min at 37 °C, followed by type IV collagenase (Sigma-Aldrich, St Louis, MO, USA) for 12–16 h (h) at 37 °C. Chondrocytes were recovered, seeded into 10 cm tissue culture plates, and maintained in a 5% CO_2_, 90% humidified atmosphere at 37 °C until 80% confluence was reached. Cells were cultured in Dulbecco’s Modified Eagle Medium (DMEM) (Gibco, Grand Island, NY, USA) supplemented with 10% fetal bovine serum (FBS), penicillin (100 U/mL), and streptomycin (100 μg/mL) (P/S) (Gibco, Grand Island, NY, USA). OA primary chondrocytes at the first passage were used for experiments to ensure preservation of their phenotypic stability.

#### 4.2.2. T/C-28a2 Chondrocytes Cell Line

The immortalized human juvenile chondrocyte cell line T/C-28a2 [77] was used.

#### 4.2.3. Cell Culture Human Articular Chondrocytes and T/C-28a2 Cell Line

Cell culture human articular chondrocytes and T/C-28a2 cell line were cultured overnight in DMEM (Gibco) supplemented with 10% FBS and P/S (Gibco) in a humidified 5% CO_2_ atmosphere at 37 °C.

Cells were plated at a density of 5 × 10^4^ cells/well in Corning^®^ CellBIND^®^ multiwell 96 plates (MW 96) (Corning, New York, NY, USA) or 1.8 × 10^5^ cells/well in MW 12 plates (Corning, New York, NY, USA) for analysis. Cells were left to equilibrate overnight in DMEM at 5% or 2% of FBS in 5% CO_2_ at 37 °C.

#### 4.2.4. Induction of Senescence

Cell senescence was induced using either 20 ng/mL interleukin-6 (IL-6; stock solution 20 µg/mL in water), as previously described [21], or etoposide (Eto; 20 µM for primary chondrocytes and 5 µM for the cell line, stock solution 10 mM in DMSO). In both conditions, senescence inducers were combined with 10 ng/mL oncostatin M (OSM; stock solution 10 µg/mL in PBS supplemented with 0.1% BSA; Sigma-Aldrich, St Louis, MO, USA). Cells were incubated with the senescence-inducing agents for 48 h. The concentrations of Eto and OSM used in this study were based on previous reports, where this combination was shown to promote a senescent phenotype through DNA damage and stress-related signaling [78].

### 4.3. Assay for Measuring Viability in Presence of AP and PA by DRAQ7^TM^ Staining

To test the effects APPA (10 µg/mL), AP (2.3 and 10 µg/mL), and PA (7.7 and 10 µg/mL) on T/C28a2 chondrocyte viability, DRAQ7^TM^ (Thermo Fisher Scientific, Waltham, MA, USA) was used. Following 24 h of different treatments, cells were trypsinized and centrifuged at 1500 rpm for 10 min. The harvested cells were suspended in 3 µM of the fluorescent dye DRAQ7^TM^ (Thermo Fisher) and incubated in the dark, at room temperature (rt) for 10 min. The samples were run on a CytoFLeX flow cytometer (Beckman Coulter Inc., Pasadena, CA, USA). At least 1 × 10^4^ cells per assay were measured. Data were analyzed using CyExpert V 2.5 software (Beckman Coulter Inc., Pasadena, CA, USA). Positive control cells were treated with 5 µM Eto plus 10 ng/µL OSM while the negative control (basal) represented cells without treatment.

### 4.4. Determination of Senotherapeutic Activity

#### 4.4.1. Senescence Determination

The percentage of senescent cells was evaluated by the quantification of SA-β-gal activity using flow cytometry and p21—*Cyclin Dependent Kinase Inhibitor 1A* (*CDKN1A*)—gene expression, which encodes for p21, an important factor for irreversible growth arrest and cell senescence.

##### Flow Cytometry

SA-β-gal activity was detected using fluorescein di-β-D-galactopyranoside (FDG; Thermo Fisher, Waltham, MA, USA) by flow cytometry. Non-fluorescent FDG is sequentially hydrolyzed by SA-β-gal, first to fluorescein monogalactoside and then to highly fluorescent fluorescein. Cell cultures were pretreated with 5 µM or 20 µM etoposide + 10 ng/mL OSM for 48 h (Sigma-Aldrich, St Louis, MO, USA) to induce cellular senescence, and then 10 nM bafilomycin A (Sigma-Aldrich, St Louis, MO, USA) for 1 h, to modulate intracellular pH, was added. These conditions were evaluated with and without 24 h drugs treatments (APPA (10 µg/mL), AP (2.3 and 10 µg/mL), and PA (7.7 and 10 µg/mL)). FDG (20 µM) was then added to the pretreatment medium. At the end of incubation, cultures were washed with PBS, resuspended by trypsinization, and analyzed immediately using a FACScalibour flow cytometer (Becton Dickinson, East Rutherford, NJ, USA). Data were acquired and analyzed with Cellquest software V5.1 (Becton Dickinson, East Rutherford, NJ, USA). Each fluorescein signal was measured on an FL1 detector and SA-β-gal activity was estimated using the median fluorescence intensity (arbitrary units) of the cell population.

##### Gene Expression by Quantitative Real-Time PCR (qRT-PCR)

RNA extraction was achieved using TRIzol^®^ (Sigma-Aldrich, St Louis, MO, USA), following the manufacturer’s protocol. In total, 0.5 µg of RNA was reverse-transcribed into cDNA using Super Script VILO (Thermo Fisher Scientific, Waltham, MA, USA), following the manufacturer’s instructions. qRT-PCR was developed in a LightCycler 480-II Instrument (Roche, Mannheim, Germany). *p21* gene expression was calculated using the Taqman Fast Advanced Master Mix (Thermo Fisher Scientific) and normalized to internal control gene 36B4 h36B4_VIC VIC-AACATCTCCCCCTTCTCCTTTGGGCT-TAMRA probe (Thermo Fisher Scientific, Waltham, MA, USA); the Hs00355782 m1 CDKN1A FAM Taqman probe (Thermo Fisher Scientific, Waltham, MA, USA) was used. Target gene expression was calculated using the comparative CT method (ΔΔCT) and normalized to a *36B4* using as housekeeping gene. Analysis of the results was carried out using Qbase+ version 2.5 software (Biogazelle, Ghent, Belgium).

### 4.5. Apoptotic Cell Determination

The susceptibility of T/C28a2 cells to apoptosis was analyzed. Cells were incubated in the presence of 5 µM Eto + 10 ng/mL OSM for 48 h (Sigma-Aldrich, St Louis, MO, USA) with and without 24 h of drugs treatments [APPA (10 µg/mL), AP (2.3 or 10 µg/mL), and PA (7.7 or 10 µg/mL)]. Cells were harvested by trypsinization and resuspended in 1 x annexin-binding buffer prior to adding 2.5 μL of Annexin-V–fluorescein isothiocyanate (FITC) and 2.5 μL of propidium iodide (PI; ImmunoStep, Salamanca, Spain). Following 15 min of incubation, cells were analyzed with a CytoFLEX flow cytometer (Beckman Coulter). Cells (1 × 10^4^) per assay were measured. Data were analyzed using CytExpert V 2.5 software (Beckman Coulter). Apoptosis was analyzed by counting cells stained with Annexin-V–FITC and/or PI. Results were expressed as a percentage of positive cells for each dye and represent the mean ± standard error of mean (SEM) of six independent experiments.

### 4.6. Senescence and Apoptotic Cell Determination at the Same Time

The percentage of T/C-28a2-positives cells for senescence and apoptotic markers 2qw evaluated. Cells were incubated as described before. The senescent cells were determined using FDG (10 µM), and the apoptotic cells were labeled with 2.5 μL of Annexin-V–phycoerythrin (PE) (ImmunoStep, Salamanca, Spain). Positive fluorescence was analyzed in a CytoFLeX flow cytometer (Beckman Coulter). Data were analyzed using CyExpert V 2.5 software (Beckman Coulter Inc.). Results were expressed as the percentage of double-positive cells (FDG and Annexin-V–PE-positive cells) and represent the mean ± standard error of the mean (SEM) of seven independent experiments.

### 4.7. Number of Cells in Cell Culture

Wells were seeded with 8 × 10^4^ T/C28a2 cells and, when 70–80% confluence was achieved, pretreated with 5 µM Eto + OSM for 72 h; the culture medium was then changed to DMEM at 2% FBS, and cells were cultured for an additional 48 h before APPA (10 µg/mL) was added for a further 24 h. The cells were then fixed in 4% paraformaldehyde (Sigma-Aldrich, St Louis, MO, USA) for 10 min at rt, followed by incubation in 0.2% Tween20 (Sigma-Aldrich, St Louis, MO, USA) for 5 min. This step was followed by 5 min incubation with the nuclear dye 2′-(4-ethoxyphenyl)-5-(4-methyl-1-piperazinyl)-2,5′-bi-1H-benzimidazole trihydrochloride (Hoechst 33258) (Sigma-Aldrich, St Louis, MO, USA). After washing with phosphate-buffered (PB) saline, cover slips were mounted on microscopy chamber slides using Prolong Gold Antifade Reagent Mountant (Thermo Fisher Scientific, Waltham, MA, USA). Fluorescence was visualized and photographed under an Olympus BX61 fluorescence microscope. All samples were analyzed in duplicate with 3–5 fields per well and mean values and standard deviations (SDs) being calculated and represented in the graph.

### 4.8. Proliferative Cells Determination by Ki-67 Analysis

Ki-67 is a protein that is found only in cells that are dividing. T/C28a2 cells were incubated as described before. Proliferating cells were determined using Ki67-FITC (10 µM) (Sigma-Aldrich, St Louis, MO, USA). The cells were fixed using 70% ethanol during 1 h at −20 °C, then centrifuged at 1500 rpm, and washed with phosphate-buffered saline (PBS), pH 7.2, and 1% bovine serum albumin (BSA, Sigma-Aldrich). Then, Ki67, prepared in the same solution, was added and cells were incubated for 20 min at rt in darkness. Positive fluorescence was analyzed by a CytoFLeX flow cytometer (Beckman Coulter). Results were expressed as a percentage of positive cells and represent the mean ± standard error of the mean (SEM) of six independent experiments.

### 4.9. Western Blotting (WB)

Total protein was obtained from 6 × 10^5^ T/C28a2 cells lysed, and then the WB was developed following the methodology described before [79]. The intensity of the bands was analyzed using Amersham Imager 600 software, and proteins on WB were quantified using Image J software V1.54n (https://imagej.nih.gov/ij/ accessed on 12 January 2025).

The WB was developed to analyze the protein levels of microtubule-associated proteins 1A/1B light chain 3B (LC3) (1:1000; Cell Signaling Technology, Beverly, MA, USA; #3868), phospho-ribosomal protein S6 (p-rpS6) (1:2000; Cell Signaling Technology, Beverly, MA, USA; #4858) and glyceraldehyde-3-phosphate dehydrogenase (GAPDH) (1:5000; Cell Signaling Technology, Beverly, MA, USA; #2118). LC3 plays a central role in autophagy while p-rpS6 is a key downstream target of mTOR and plays a significant role in regulating protein synthesis and cell growth.

### 4.10. Statistical Analyses

Data are presented as mean ± standard error of the mean (SEM) from at least four independent experiments, unless otherwise indicated. Statistical analyses were performed using GraphPad Prism version 8.0.0 (GraphPad Software, La Jolla, CA, USA). Group comparisons were assessed using the non-parametric unpaired Mann–Whitney U test. Statistical significance was defined as *p* ≤ 0.05. Asterisks (*) indicate comparisons versus the basal (untreated) condition; hash symbols (#) indicate comparisons versus the 5 µM etoposide + 10 ng/mL oncostatin M (Eto + OSM) condition.

## 5. Conclusions

This report indicates that APPA potentially has senotherapeutic activity against senescent human chondrocytes. Reductions in SA-β-gal activity with an increase in cell numbers and the proliferation marker ki67 suggest possible senomorphic effects, whereas reductions in SA-β-Gal and an increase in apoptosis indicate senolytic activity. These effects were only seen with the APPA combination, not the individual components. Recent evidence supports the concept that the distinction between senolytic and senomorphic drugs is artificial and has been labeled ‘fuzzy’ [75]. Additional experiments are needed to further explore the effects reported here with APPA and to thoroughly characterize both chondrocyte populations.

## Figures and Tables

**Figure 1 pharmaceuticals-18-01386-f001:**
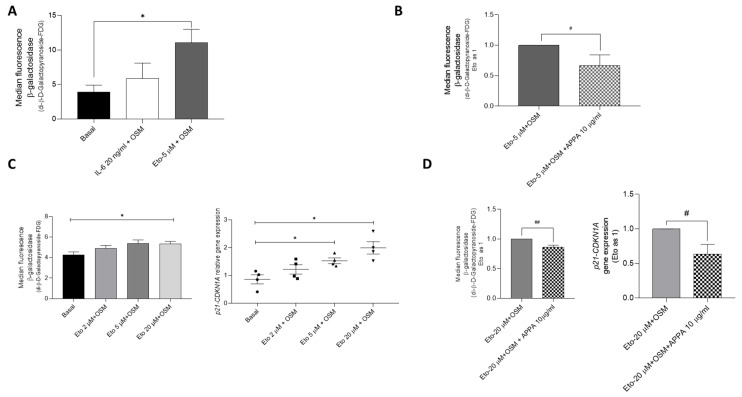
**Optimization of the stimulus to induce senescence.** (**A**) Optimization of positive stimuli in T/C-28a2 was evaluated using flow cytometry-determined SA-β-galactosidase activity. Cells were incubated for 48 h in presence of either 20 ng/mL IL-6 or 5 µM Eto, both in combination with 10 ng/mL OSM. (**B**) Senescent cells evaluated by flow cytometry were represented in the presence of 5 µM Eto + 10 ng/mL OSM ± 10 µg/mL APPA (24 h). (**C**) Optimization of the concentration of etoposide (2, 5 and 20 µM) in combination with 10 ng/mL OSM in primary human chondrocyte culture for 48 h, evaluated using flow cytometry and p21 (*CDKN1A*) gene expression. Data were obtained from four independent donors. (**D**) Senescent primary human chondrocyte in the presence of positive stimuli ± 10 µg/mL APPA (for 24 h) was evaluated by flow cytometry and gene expression levels of p21 (*CDKN1A*). Data were obtained from six independent donors. Values are presented as mean ± SEM and analyzed by Mann–Whitney test; * relative to basal condition (cells without treatment); #, ## relative to Eto + OSM; (*, # *p* ≤ 0.05, ## *p* ≤ 0.005). Eto = etoposide; OSM = oncostatin M.

**Figure 2 pharmaceuticals-18-01386-f002:**
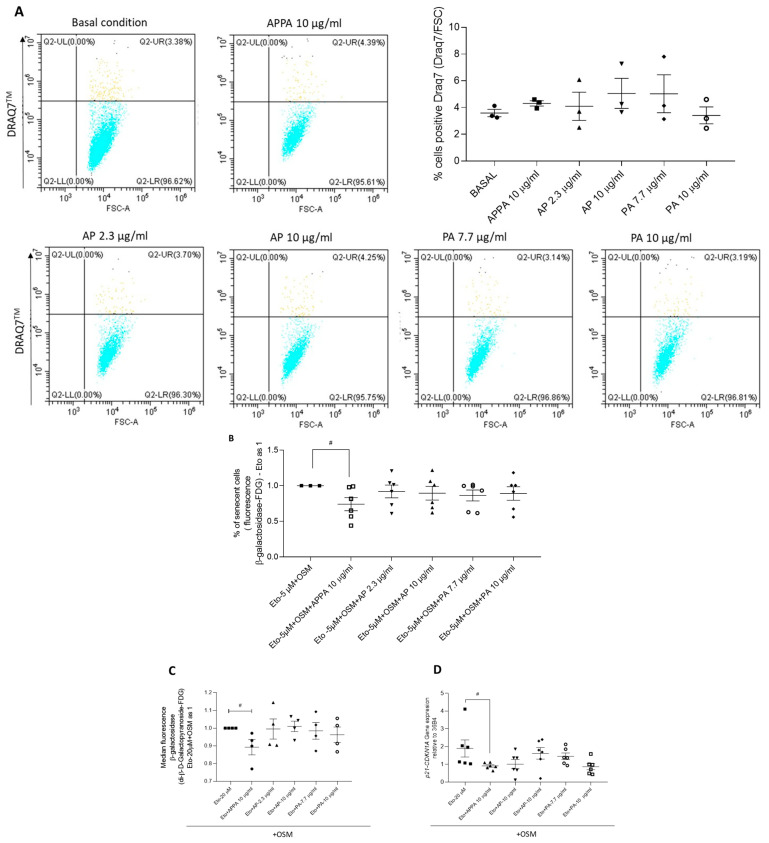
**Effect of AP and PA on cell viability and senescence.** (**A**) Representative dot plots for T/C28a2 cells treated with APPA, AP, and PA for the DRAQ7 viability assay. Untreated cells (basal condition), cells treated with APPA 10 µg/mL in the upper line, and cells incubated with AP and PA in the bottom line. Plots show cells in each region, and dead cells are defined by DRAQ7 positivity (UR region) (blue; cells negative DRAQ7 and light yellow cells positive DRAQ7). The quantification of percentages of cells positive for DRAQ7 is represented. Data are represented as the mean ± standard error of the mean (SEM). All data were obtained from three independent experiments performed with two replicates. (**B**) T/C-28a2 human chondrocytes treated with 5 µM Eto + 10 ng/µL OSM (Eto + OSM) and then APPA 10 µg/mL, AP 2.3 10 µg/mL, or PA 7.7 10 µg/mL were added, and senescent cells were evaluated by FDG. (**C**,**D**) Human primary chondrocytes treated with Eto (20 µM) + OSM 10 ng/mL and then APPA 10 µg/mL, AP 2.3 10 µg/mL, or PA 7.7 10 µg/mL were added, and senescent cells were evaluated by FDG (**C**,**D**) gene expression of p21 (*CDKN1*). Data were obtained from five independent donor experiments performed with two replicates. Values are presented as mean ± SEM and analyzed by the Mann–Whitney test (# *p* ≤ 0.05). # relative to Eto + OSM.

**Figure 3 pharmaceuticals-18-01386-f003:**
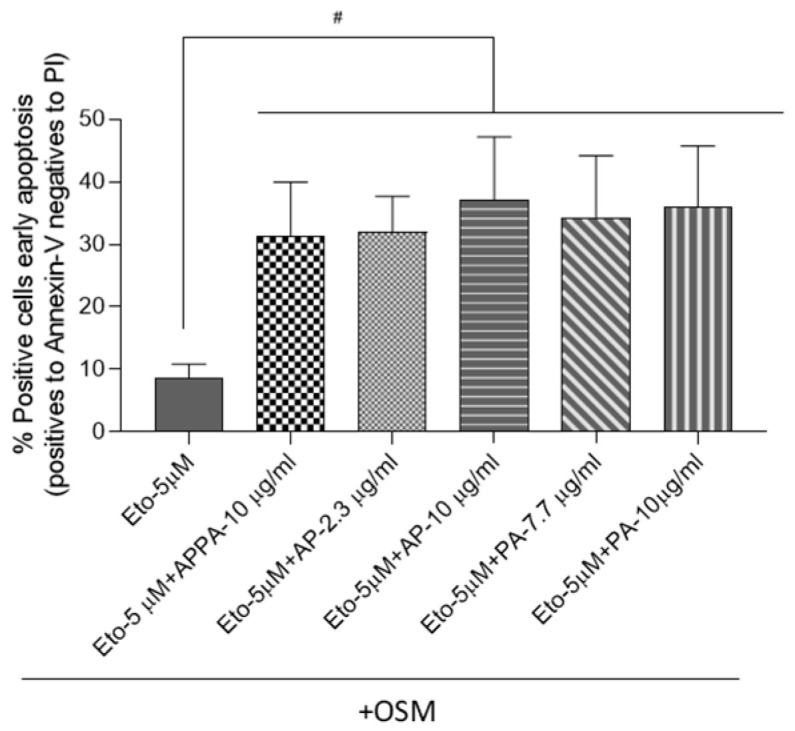
**APPA increased the apoptotic cells in senescence environments.** T/C-28a2 human chondrocytes treated with 5 µM Eto + 10 ng/µL OSM (Eto) for 48 h and then APPA 10 µg/mL, AP (2.3 and 10 µg/mL), and PA (7.7 and 10 µg/mL) were added. Analysis of apoptotic cells was undertaken by flow cytometry; data represent early apoptosis (cells positive for Annexin-V and negative for PI). All data were obtained from six independent experiments performed with two replicates. Data are represented as mean ± SEM and analyzed by the unpaired Mann–Whitney test (# *p* < 0.05). # relative to Eto (5 µM Eto + 10 ng/µL OSM).

**Figure 4 pharmaceuticals-18-01386-f004:**
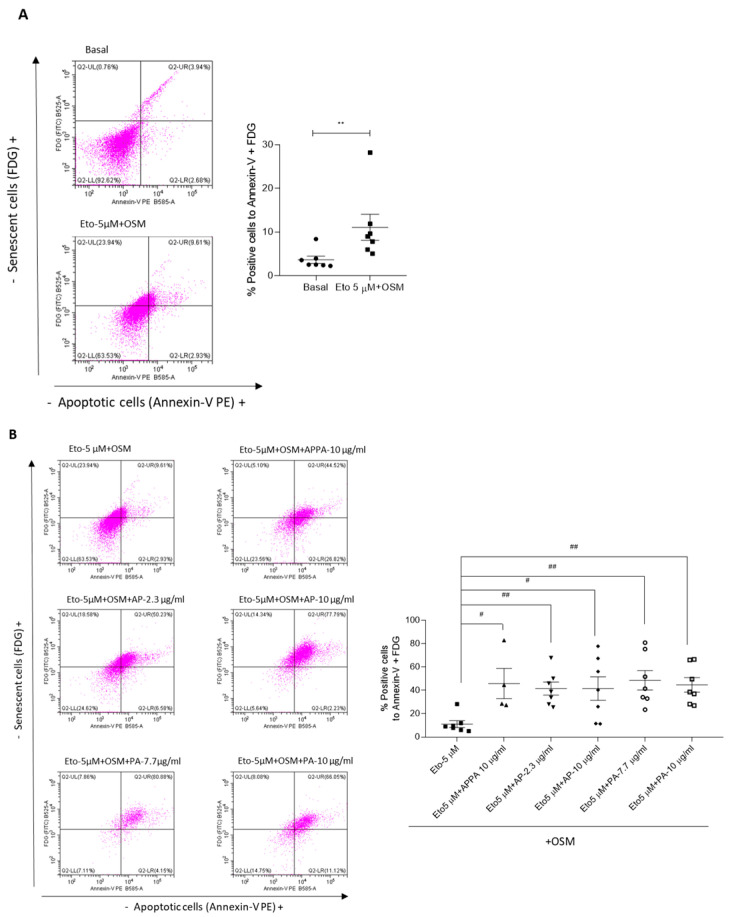
**APPA increased apoptosis in senescent chondrocytes.** Senescence (FDG)/apoptosis (Annexin-V) double staining was performed on T/C-28a2 cells treated with 5 µM etoposide + 10 ng/µL OSM during 48 h and then APPA 10 µg/mL, AP (2.3 and 10 µg/mL), and PA (7.7 and 10 µg/mL) for 24 h. (**A**) Dot plot for cells in basal condition and treated with 5 µM etoposide + 10 ng/µL OSM; the graph shows the cells with double staining. (**B**) Representative dot plot of the percentages of FDG/Annexin-V double positives (upper right quadrant_UR_). The graphs are representative of four duplicate experiments. FDG (FITC)/Annexin-V (PE) double-staining flow cytometry analysis of cell senescence and apoptotosis. UL: senescent cells; UR: senescent and apoptotic cells; LL: normal cells (negative cells); LR: apoptotic cells. Data are represented as mean ± SEM and analyzed by the unpaired Mann–Whitney test (# *p* < 0.05, **, ## *p* < 0.01).). **, relative to Basal condition; #,## relative to 5 µM etoposide + OSM.

**Figure 5 pharmaceuticals-18-01386-f005:**
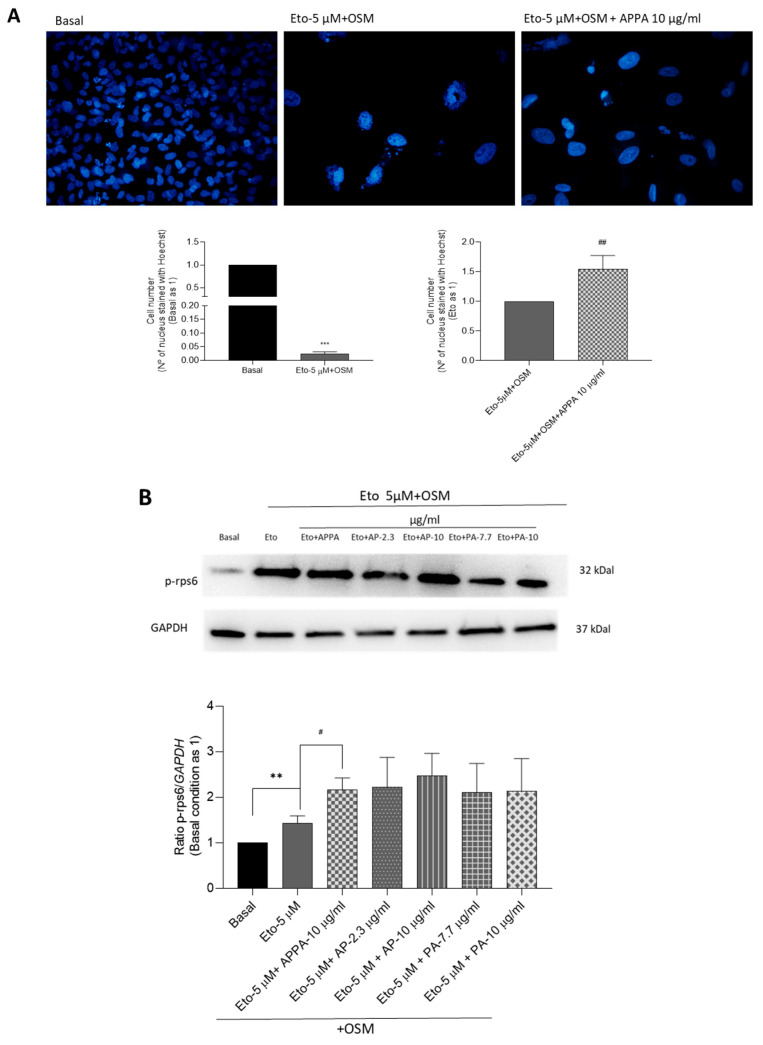
**APPA effect on cell proliferation.** (**A**) T/C-28a2 human chondrocytes treated with Eto (5 µM) + OSM (10 ng/µL) for 48 h and then 10 µg/mL APPA for 24 h. Representative image of Hoechst 33342 staining and quantification of cell number relative to basal (equal to 1) or to Eto (equal to 1) groups. All images were acquired using a 10× objective. (**B**) Western blotting of protein extracts probed with antibody specific for phospho-ribosomal protein S6 (p-rpS6) and GAPDH. Representative blots are shown, along with numeric data obtained by densitometry. (**C**) Flow cytometry plots of Ki67expression. The percentage of Ki67-positive cells wwas measured and analyzed in each population (P1 and P2). All data were obtained from six independent experiments performed with two replicates. Values are presented as mean ± SEM and analyzed by the Mann–Whitney test (# *p* ≤ 0.05; **, ## *p* ≤ 0.01; ***, ### *p* ≤ 0.0005). **, *** relative to basal condition; #, ##, ### relative to Eto + OSM.

## Data Availability

Data are contained within the article or Appendix A; the original contributions presented in this study are included in the article/Appendix A. Further inquiries can be directed at the corresponding authors.

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
