# Peer review of "The Senotherapeutic Effects of APPA (Apocynin [AP] and Paeonol [PA]) on Senescent Human Chondrocytes"

_pharmaceuticals, 2025, doi:10.3390/ph18091386_

Round 1

Reviewer 1 Report

Comments and Suggestions for Authors

This study investigates the effects of APPA on senescent human chondrocytes. Cellular senescence is believed to play a pivotal role in several age-related diseases, due to the detrimental influence of senescent cells on the surrounding microenvironment. Therefore, studies aimed at counteracting or reducing the senescent phenotype are of great importance and should be supported. The study is very interesting; however, it contains several technical inaccuracies that should be addressed before it can be positively considered for publication.

Introduction:

  • The aim of the study is presented only in the last sentence of the Introduction. I suggest stating the aim earlier, strengthening its connection with cellular senescence, the current lack of effective senotherapeutic agents, and the advantages of using APPA.

Results:

  • Paragraph 2.1: The section is overly confusing. Although the treatment details are described in the Materials and Methods section, I recommend clarifying the concentrations used for senescence induction and APPA treatment, as well as the treatment durations.
  • What is the rationale for using a combination of OSM and Eto? Did you test prolonged Eto treatment alone to achieve the same results?
  • The β-galactosidase assay is not an adequate test to confirm cellular senescence. All xenobiotic agents induce a lysosomal activity-based cellular response to remove damaged organelles; hence, β-galactosidase activity is naturally elevated in the presence of xenobiotics. This is a normal cellular reaction, not an indication of stable senescence. To confirm stable cellular senescence, multiple assays should be used, such as p21 and p16 protein expression analysis, cell cycle assays, phagocytosis activity, and SASP factor secretion.
  • Real-time PCR for p21 expression is insufficient. You should include Western blot data for both p21 and p16 protein expression, as these are essential to confirm a fully established senescent state.
  • In the legend of Figure 1C, you state that experiments were repeated using chondrocytes from 4 donors. Why were only 4 donors tested when 18 donors are reported in the Materials and Methods, paragraph 4.2.1?
  • Legend of Figure 1D: “senescent cells in the presence of …” — which cells are you referring to? Primary chondrocytes or T/C-28a2?
  • How do you explain the differences in response between T/C-28a2 and primary human chondrocytes when using varying concentrations of senescence-inducing agents?
  • Paragraph 2.2: Please specify the treatment durations for each cell type used.
  • How do you explain the differences in cellular responses when using APPA compared to its individual components?
  • Paragraph 2.4: To better assess the effects of APPA and its individual components on cell proliferation, it is important to include the baseline proliferation rates of T/C-28a2 cells and primary human chondrocytes under untreated conditions.
  • Figure 5A: Fluorescence images lack a scale bar. The magnification of basal samples appears lower than that of treated samples, and nuclei sizes are inconsistent. Please replace these images with ones more representative of the data.
  • Regarding the Ki67-positive cell data, how do you explain the reduced proliferation after Eto+APPA treatment? This seems contradictory to previous results, as the Eto+APPA combination appears to increase cellular senescence.
  • Overall, the superiority of APPA compared to its individual components is unclear. Several results appear very similar, and the statistical analysis performed seems weak.

Materials and Methods:

  • Paragraph 4.2.1: Please provide more detailed information on the procedure for isolating human chondrocytes from OA hip cartilage.
  • Given that you have access to OA human chondrocytes, it is unclear why most experiments were conducted on the T/C-28a2 chondrocyte cell line rather than on OA human chondrocytes. These models differ substantially, and the results and conclusions could vary significantly. Moreover, OA human chondrocytes better represent the pathological and senescent environment of OA.
  • Paragraph 4.2.4: Although you reference previous publications for your senescence induction protocol, I recommend expanding this section to include more details on concentrations, treatment durations, and the solvent used (e.g., water, DMSO)

Author Response

Dear Reviewer-1

We thank the reviewer for the positive assessment of the relevance and interest of our study. We greatly appreciate the recognition of the importance of targeting cellular senescence in age-related diseases, as well as the constructive feedback provided. We agree that addressing the technical issues raised will strengthen the manuscript, and we have carefully revised the text accordingly. Below, we provide detailed responses to each of the points.

Introduction:

  • The aim of the study is presented only in the last sentence of the Introduction. I suggest stating the aim earlier, strengthening its connection with cellular senescence, the current lack of effective senotherapeutic agents, and the advantages of using APPA.

Response. Thank you for your comment. Following your suggestion, we have restructured the Introduction to present the aim of the study earlier and more clearly. The revised section now emphasizes the relevance of cellular senescence in osteoarthritis, the current lack of effective senotherapeutic agents, and the potential advantages of using APPA. This reorganization ensures that the study rationale and objectives are immediately clear to the reader.

Results:

  • Paragraph 2.1: The section is overly confusing. Although the treatment details are described in the Materials and Methods section, I recommend clarifying the concentrations used for senescence induction and APPA treatment, as well as the treatment durations.

Response. Thank you for your suggestion. To improve clarity, we have revised the Results section to explicitly indicate the concentrations and treatment durations. The section now reads: "Data showed that when T/C-28a2 cells were incubated with either interleukin-6 (IL-6, 20 ng/ml) or Etoposide (Eto, 5 µM), both in the presence of oncostatin M (OSM, 10 ng/ml), the number of senescent cells increased; however, this increase was significant only for Eto+OSM (Figure 1A). Therefore, 5 µM Eto combined with 10 ng/ml OSM (Eto+OSM) for 48 hours was selected for subsequent experiments with the T/C-28a2 cell line. When senescent T/C-28a2 cells induced by Eto+OSM were treated with APPA for 24 hours,…

We have also updated the legend of Figure 1 to reflect these concentrations and treatment durations for improved clarity.

  • What is the rationale for using a combination of OSM and Eto? Did you test prolonged Eto treatment alone to achieve the same results?

Response: Thank you for your comment. In our study, senescence in human chondrocytes was induced pharmacologically using a combination of etoposide (Eto at 20 µM for primary chondrocytes or 5 µM in the case of the cell line) and oncostatin M (OSM, 10 ng/mL). This approach has been previously reported to induce a senescent phenotype in chondrocytes and other cell types by promoting DNA damage and stress-related signaling, rather than by replicative senescence, oxidative stress (e.g., Hâ‚‚Oâ‚‚), or irradiation. While treatment with Eto alone can also induce senescence, extending the incubation time substantially leads to irreversible apoptosis, reducing cell viability and limiting the utility of the model. The combination with OSM allows for a robust induction of senescence while maintaining sufficient cell survival for downstream analyses. Georget et al., 2023 (PMID: 37659108) described that Etoposide-induced senescence model may help investigate the initiation of cellular senescence in chondrocytes, and provide a useful model to develop therapeutic approaches to target senescence in OA

  • The β-galactosidase assay is not an adequate test to confirm cellular senescence. All xenobiotic agents induce a lysosomal activity-based cellular response to remove damaged organelles; hence, β-galactosidase activity is naturally elevated in the presence of xenobiotics. This is a normal cellular reaction, not an indication of stable senescence. To confirm stable cellular senescence, multiple assays should be used, such as p21 and p16 protein expression analysis, cell cycle assays, phagocytosis activity, and SASP factor secretion.

Response. Thank you for your comment. In this study, SA-β-Gal activity was assessed using a quantitative flow cytometry-based method with fluorescein di-β-D-galactopyranoside (FDG), which allows precise measurement of senescence-associated β-galactosidase activity at pH 6.0 while controlling for lysosomal background activity. Cells were pretreated with Eto+OSM to induce senescence and bafilomycin A was used to modulate intracellular pH, ensuring specificity of the SA-β-Gal signal. FDG fluorescence was measured on flow cytometer and analyzed using median fluorescence intensity, providing a robust, quantitative readout rather than relying solely on qualitative staining. Complementing this, p21 expression by real-time PCR was significantly upregulated, supporting senescence induction.

We acknowledge that additional complementary assays, including p16/protein-level analysis, cell cycle profiling, and SASP evaluation, would further strengthen confirmation of stable senescence. These analyses are planned for future studies. Taken together, the combination of quantitative SA-β-Gal flow cytometry, p21 expression, and functional cellular responses supports the presence of senescence in our model and demonstrates the senotherapeutic activity of APPA. In addition, experiments where IL-1ẞ was used as an inflammatory induction agent, which is also known to induce senescence in chondrocytes (PMID: 24572376), APPA was shown to reduce the expression of known SASP markers such as TNFα and MMPs 13 and 3 (PMID: 38256951).

To improve the understanding of our data, we have included the following text in the Discussion section: “We acknowledge that additional complementary assays, including p16/protein-level analysis, cell cycle profiling, and SASP biomarker evaluation and single cell techniques, would further strengthen confirmation of stable senescence.”

  • Real-time PCR for p21 expression is insufficient. You should include Western blot data for both p21 and p16 protein expression, as these are essential to confirm a fully established senescent state.

Response. Thank you for your suggestion. In the present study, we assessed p21 expression by real-time PCR in human primary chondrocytes, which provided clear evidence of senescence induction under our experimental conditions (Eto+OSM). We acknowledge that including Western blot analyses for both p21 and p16 would further strengthen the confirmation of a fully established senescent state. We will consider incorporating these protein-level analyses in future studies to expand and validate our findings.

  • In the legend of Figure 1C, you state that experiments were repeated using chondrocytes from 4 donors. Why were only 4 donors tested when 18 donors are reported in the Materials and Methods, paragraph 4.2.1?

Response. Thank you for your question. The reason that only 4 of the 18 donors are referenced in panel 1C is that, given the age and advanced stage of OA of all donors, the number of chondrocytes obtained is limited. Since the experiments are performed at passage S1 to ensure that the cells maintain a chondrocyte phenotype rather than becoming chondrocyte-like, the cell numbers are typically insufficient to perform more than one or two experiments per donor.

Furthermore, experiments with primary chondrocytes are performed sequentially, meaning the chondrocytes from each donor are used in the order in which the samples arrive at the laboratory each week and they are ready to carry out the experiment. This is how we perform the different experiments. We do not perform experiments with a pool of chondrocytes in which cells from different donors are mixed. In this case, four donors will be used to perform four experiments. Other donors were used to carry out other experiments showed in other figures

  • Legend of Figure 1D: “senescent cells in the presence of …” — which cells are you referring to? Primary chondrocytes or T/C-28a2?

Response. Thank you for your question. Figure 1D refers to primary human chondrocytes. To avoid any ambiguity, we have revised the legend to:D. Senescent primary human chondrocytes in the presence of positive stimuli….”

  • How do you explain the differences in response between T/C-28a2 and primary human chondrocytes when using varying concentrations of senescence-inducing agents?

Response: Thank you for your comment. Differences in response between T/C-28a2 cells and primary human chondrocytes likely reflect intrinsic differences between immortalized cell lines and primary cells. T/C-28a2 cells, being immortalized, may have altered cell cycle regulation, DNA damage responses, and stress signaling pathways compared with primary chondrocytes, which can result in different sensitivity to senescence-inducing agents. In primary human chondrocytes, responses to senescence-inducing stimuli are further influenced by donor-specific factors, including age and inherent biological variability, making it necessary to use higher concentrations or prolonged exposure to achieve comparable senescence-associated phenotypic changes across all donors evaluated. Primary chondrocytes have limited ability to proliferate and are known to de-differentiate in culture so the cell line is likely to be a more reliable and reproducible option for initial exploratory studies. Importantly in our studies the results seen in experiments with primary chondrocytes were largely similar to those observed in the cell line. It has also been reported recently that there are several subtypes of chondrocytes in human cartilage which potentially would add an additional element of variability in response to induction of senescence and response to treatments. (PMID: 35767667, PMID: 36647698).

Paragraph 2.2: Please specify the treatment durations for each cell type used.

Response: Following your suggestion, we have specified the treatment duration for each cell type. The section now reads: “A dot plot from the DRAQ7 assay was used to estimate cell viability in the presence of both drug concentrations during 24 h of incubation.

  • How do you explain the differences in cellular responses when using APPA compared to its individual components?

Response: Thank you for your question. The combination of apocynin (AP) and paeonol (PA) in APPA provides broader and more potent modulation of key inflammatory and stress-related pathways than either compound alone, enhancing both anti-inflammatory and cytoprotective effects (PMID: 3238062, PMID: 38256951). AP primarily inhibits NADPH oxidase–dependent reactive oxygen species generation and modulates NF-κB signaling, while PA inhibits NF-κB activation, reduces pro-inflammatory cytokine expression, and upregulates Nrf2 signaling. By acting through partially complementary mechanisms, the combination achieves a more robust effect than either component individually.

  • Paragraph 2.4: To better assess the effects of APPA and its individual components on cell proliferation, it is important to include the baseline proliferation rates of T/C-28a2 cells and primary human chondrocytes under untreated conditions.

Response: Thank you for your comment. We agree that including baseline proliferation rates is important for interpreting the effects of APPA and its individual components. In our analyses, the basal (untreated) condition is included to facilitate the assessment of proliferation changes. Comparisons were made between the basal condition and Eto-treated cells, and between Eto-treated cells and those treated with Eto plus AP, PA, or APPA. While baseline proliferation was not assessed in primary chondrocytes in this study, it would be very interesting to perform such analyses in the future using cells from younger donors, ideally within a defined age range, to further explore age-dependent differences in proliferative responses.

  • Figure 5A: Fluorescence images lack a scale bar. The magnification of basal samples appears lower than that of treated samples, and nuclei sizes are inconsistent. Please replace these images with ones more representative of the data.

Response. Thank you for your comment. We have included in the figure legend that all images were acquired using a 10× objective. Regarding the reviewer’s comment on nuclear morphology, we acknowledge that some variability is present. This disparity may be related to the different senescence states described in our study: while some cells are undergoing senescence, others may be entering apoptosis, and a subset may still be proliferating. We believe this biological heterogeneity is intrinsic to the process under investigation and is now better represented in the updated images.

  • Regarding the Ki67-positive cell data, how do you explain the reduced proliferation after Eto+APPA treatment? This seems contradictory to previous results, as the Eto+APPA combination appears to increase cellular senescence.

Response. Thank you for your comment. The Ki67 data were analyzed by distinguishing two populations of T/C-28a2 chondrocytes (P1 and P2) identified in the dot-plots (Figure 5C), which showed clearly different behaviors under the experimental conditions. In population P1, the percentage of Ki67-positive cells decreased in the presence of Eto+OSM compared with basal conditions, and this reduction was even more pronounced with Eto+OSM+APPA. In contrast, population P2—which was almost absent under basal conditions but increased after Eto+OSM treatment—showed a significant rise in Ki67-positive cells when APPA was added, consistent with enhanced proliferative activity.

These apparently divergent responses suggest that APPA acts differentially depending on the cellular state. A fraction of cells (likely more senescent) responds by enhancing apoptotic pathways, which is consistent with senolytic activity, whereas another subset (likely less damaged or in a pre-senescent state) responds by recovering proliferative potential, which is more consistent with senomorphic effects. Supporting this interpretation, Eto+OSM+APPA treatment increased total cell numbers and p-rps6 levels, in line with proliferative recovery under a senescence-inducing environment.

Taken together, these data indicate that the reduced proliferation observed in one subpopulation does not contradict our overall findings but rather reflects heterogeneous cellular responses to Eto+OSM and APPA. This dual behavior reinforces the conclusion that APPA exerts both senolytic and senomorphic effects, best described under the broader term senotherapeutic activity, in agreement with recent perspectives suggesting that the distinction between these two activities is “fuzzy” (PMID: 35781578).

To improve the understanding of our data, we have included the following text in the Results section: “In summary of these data, Ki67 distinguished two populations of T/C-28a2 chondrocytes (P1 and P2), which showed clearly different behaviors under the experimental conditions. In population P1, the percentage of Ki67-positive cells decreased in the presence of Eto+OSM compared with basal conditions, and this reduction was even more pronounced with Eto+OSM+APPA. In contrast, population P2—which was almost absent under basal conditions but increased after Eto+OSM treatment—showed a significant rise in Ki67-positive cells when APPA was added, consistent with enhanced proliferative activity.”

  • Overall, the superiority of APPA compared to its individual components is unclear. Several results appear very similar, and the statistical analysis performed seems weak.

Response. Thank you for your comment. The rationale for combining apocynin (AP) and paeonol (PA) in a fixed 2:7 ratio (APPA) is supported by both mechanistic and empirical evidence. AP and PA are plant-derived phenolic compounds with complementary anti-inflammatory and immunoregulatory activities: AP inhibits NADPH oxidase–dependent reactive oxygen species generation and modulates NF-κẞ signaling, whereas PA inhibits NF-κẞ activation, reduces pro-inflammatory cytokine expression, and enhances Nrf2 signaling. The combination of AP and PA in APPA provides broader and more potent modulation of these pathways than either compound alone, enhancing anti-inflammatory and cytoprotective effects (PMID: 3238062, 38256951).

The 2:7 AP:PA ratio was established in preclinical studies as the most effective proportion and subsequently validated in a Phase 2a clinical trial in patients with knee osteoarthritis (PMID: 38697511). Animal studies and n=1 studies in dogs with naturally occurring OA also demonstrated that this ratio produced optimal therapeutic effects, guiding its selection for clinical testing. The efficacy of this ratio has been confirmed in dog and human studies

In our current study, APPA at the 2:7 ratio consistently showed stronger and more reproducible effects on senescence-related endpoints in human chondrocytes, including reduction of SA-β-Gal activity, modulation of Ki67-positive cell populations, apoptosis induction in senescent cells, and increased p-rps6 levels, whereas the individual components AP and PA did not produce significant effects on these parameters. These findings demonstrate that the combination is necessary to achieve the observed senotherapeutic effects and highlight the superior efficacy of APPA compared with its individual components.

To improve the understanding of our data, we have included the following text in the Discussion section “The combination of AP and PA in APPA has been shown to provide broader and more potent modulation of inflammatory and immunoregulatory pathways than either compound alone”

Materials and Methods:

  • Paragraph 4.2.1: Please provide more detailed information on the procedure for isolating human chondrocytes from OA hip cartilage.

Response. Thank you for your recommendation. Following your suggestion, we have expanded the Methods section to provide more detailed information on the isolation of human chondrocytes from OA hip cartilage: “The chondrocytes were isolated after surgery. Briefly, cartilage sections were aseptically removed from each donor and enzymatically digested with trypsin (Sigma-Aldrich) for 15 minutes at 37 °C, followed by type IV collagenase (Sigma-Aldrich) for 1216 hours at 37 °C. Chondrocytes were recovered, seeded into 10-cm tissue culture plates, and maintained in a 5% COâ‚‚, 90% humidified atmosphere at 37 °C until 80% confluence was reached. Cells were cultured in Dulbecco's Modified Eagle Medium (DMEM) (Gibco, Grand Island, NY, USA) supplemented with 10% fetal bovine serum (FBS), penicillin (100 U/ml), and streptomycin (100 μg/ml) (Gibco). OA primary chondrocytes at the first passage were used for experiments to ensure preservation of their phenotypic stability.”

  • Given that you have access to OA human chondrocytes, it is unclear why most experiments were conducted on the T/C-28a2 chondrocyte cell line rather than on OA human chondrocytes. These models differ substantially, and the results and conclusions could vary significantly. Moreover, OA human chondrocytes better represent the pathological and senescent environment of OA.

Response: Thank you for your comment. We agree that OA human chondrocytes more closely represent the pathological and senescent environment of osteoarthritis. In this study, we used T/C-28a2 cells for most experiments because they provide a higher number of cells as well as a more consistent and reproducible model, allowing us to optimize experimental conditions and reduce variability inherent to primary cells from multiple donors. Importantly, key findings were validated in primary OA chondrocytes whenever feasible, confirming that the observed effects of AP, PA, and APPA are not restricted to the cell line. Future studies including a larger number of primary donors, ideally across a defined age range, will help to further corroborate these results and assess potential donor-dependent variability.

  • Paragraph 4.2.4: Although you reference previous publications for your senescence induction protocol, I recommend expanding this section to include more details on concentrations, treatment durations, and the solvent used (e.g., water, DMSO)

Response. Thank you for your comment. In accordance with the suggestion, we have expanded the Methods section to provide additional details of the senescence induction protocol. Specifically, we now include the precise concentrations, treatment durations, and the solvents used for each compound (IL-6 dissolved in water, Etoposide in DMSO, and OSM in PBS supplemented with 0.1% BSA). These details have been incorporated to ensure clarity and reproducibility.

To improve the understanding of our data, we have included the following text in the Methods section: “Cell senescence was induced using either 20 ng/ml Interleukin-6 (IL-6; stock solution 20 µg/ml in water, as previously described [21]) or Etoposide (Eto; 20 µM for primary chondrocytes and 5 µM for the cell line, stock solution 10 mM in DMSO). In both conditions, senescence inducers were combined with 10 ng/ml Oncostatin M (OSM; stock solution 10 µg/ml in PBS supplemented with 0.1% BSA; Sigma-Aldrich). Cells were incubated with the senescence-inducing agents for 48 h

Reviewer 2 Report

Comments and Suggestions for Authors

Suggested Revisions:

The manuscript is overall well-conceived, and the experimental design appears comprehensive, covering senescence biomarkers, oxidative stress assays, and ECM-related markers. However, there are areas where methodological rigor, mechanistic insights, and data interpretation need to be strengthened.

1.      The rationale for combining AP and PA requires stronger mechanistic justification. Are their molecular targets complementary or overlapping? Without a clear hypothesis-driven explanation, the combination might appear arbitrary?

2.      The method of inducing senescence in human chondrocytes must be more clearly described. Was replicative senescence, oxidative stress (e.g., Hâ‚‚Oâ‚‚), or irradiation used? 

3.      Some of the claims regarding “reversal of senescence” may be overstated. The presented evidence is more consistent with attenuation of senescence-associated phenotypes (senomorphic activity), rather than clearance of senescent cells (senolytic activity).   The authors should carefully distinguish between “senolytic” and “senomorphic” terminology?

4.      The manuscript briefly mentions osteoarthritis implications, but this needs to be developed. How could APPA be delivered in vivo? Would intra-articular injection or oral administration be feasible?

5.      Are there any prior reports of AP or PA in preclinical OA models that can be referenced?

6.      Pharmacokinetics, bioavailability, and potential toxicity in humans should be at least mentioned?

7.      In the results section, avoid subjective language like “remarkably reduced” or “dramatically increased”; use quantitative descriptions?

8.      The introduction could better integrate current literature on senotherapeutics in OA (e.g., studies with dasatinib + quercetin, fisetin, or navitoclax)?

9.      Ensure consistency in abbreviations (e.g., APPA, SA-β-gal, ECM)?

Author Response

Dear Reviewer-2

We sincerely thank the reviewer for the careful evaluation of our manuscript and the constructive feedback provided. We appreciate the recognition of the overall design and scope of our study, and we agree that addressing the points raised will help us to strengthen the methodological rigor, mechanistic interpretation, and clarity of our work.

Below, we provide detailed responses to each of the comments.

  1. The rationale for combining AP and PA requires stronger mechanistic justification. Are their molecular targets complementary or overlapping? Without a clear hypothesis-driven explanation, the combination might appear arbitrary?

Response. Thank you for your comment. The rationale for combining apocynin (AP) and paeonol (PA) in a fixed 2:7 ratio (APPA) is supported by preclinical and clinical evidence. Both AP and PA are compounds with demonstrated anti-inflammatory and immunoregulatory properties, but they act through partially complementary mechanisms. AP is known to inhibit NADPH oxidase–dependent reactive oxygen species generation and to modulate NF-κB signaling, while PA has been shown to inhibit NF-κB activation, reduce pro-inflammatory cytokine expression, and enhance Nrf2 signaling.

APPA is a synergistic combination of two bioactive phenolic isomers, apocynin (4-hydroxy-3-methoxyacetophenone) (AP) and paeonol (2'-hydroxy-4'-methoxyacetophenone) (PA) which was developed by a series of n=1 studies in dogs with naturally occurring OA undertaken by one of the authors (NL) (This sentence was introduced in the introduction section). In the Material and Method section: “The 2:7 AP:PA ratio was established in preclinical studies as the most effective proportion and subsequently validated in a Phase 2a clinical trial in patients with knee osteoarthritis (51, 52, 55). Animal studies and n=1 studies in dogs with naturally occurring OA also demonstrated that this ratio produced op-timal therapeutic effects, guiding its selection for clinical testing”.

The combination of AP and PA in APPA has been shown to provide broader and more potent modulation of these pathways than either compound alone, thereby enhancing anti-inflammatory and cytoprotective effects (PMID: 3238062, PMID: 38256951). Importantly, the 2:7 ratio was established in preclinical studies as the most effective proportion, and its efficacy and safety have since been validated in a Phase 2a clinical trial in patients with knee OA (PMID: 38697511).

Thus, the combination is not arbitrary, but rather based on mechanistic complementarity and empirical evidence demonstrating superior efficacy of APPA compared with its individual components.

  1. The method of inducing senescence in human chondrocytes must be more clearly described. Was replicative senescence, oxidative stress (e.g., H2O2), or irradiation used?.

Response. Thank you for your comment. In our study, senescence in human chondrocytes was induced pharmacologically using a combination of etoposide (Eto at 20 µM for primary chondrocytes or 5 µM in the case of the cell line) and oncostatin M (OSM, 10 ng/mL). This approach has been previously reported to induce a senescent phenotype in chondrocytes and other cell types by promoting DNA damage and stress-related signaling, rather than by replicative senescence, oxidative stress (e.g., Hâ‚‚Oâ‚‚), or irradiation. Georget et al., 2023 (PMID: 37659108) described that Etoposide-induced senescence model may help investigate the initiation of cellular senescence in chondrocytes, and provide a useful model to develop therapeutic approaches to target senescence in OA

We have revised the Methods section to clarify this point: “Cell senescence was induced with either 20 ng/ml Interleukin-6 (IL-6) using the con-centration described previously Cell senescence was induced using either 20 ng/ml Inter-leukin-6 (IL-6; stock solution 20 µg/ml in water, as previously described (21), or Etoposide (Eto) at 20 µM for primary chondrocytes or 5 µM in the case of the cell line. In both cases the senescence inductors were combined with 10 ng/ml of Oncostatin M (OSM, (Sig-ma-Aldrich)). or Etoposide (Eto; 20 µM for primary chondrocytes and 5 µM for the cell line, stock solution 10 mM in DMSO). In both conditions, senescence inducers were combined with 10 ng/ml Oncostatin M (OSM; stock solution 10 µg/ml in PBS supplemented with 0.1% BSA; Sigma-Aldrich). Cells were incubated with the senescence-inducing agents for 48 h. The concentrations of Eto and OSM used in this study were based on pre-vious reports, where this combination was shown to promote a senescent phenotype through DNA damage and stress-related signaling.evaluated in this work were described in previous studies (79)”.

  1. Some of the claims regarding “reversal of senescence” may be overstated. The presented evidence is more consistent with attenuation of senescence-associated phenotypes (senomorphic activity), rather than clearance of senescent cells (senolytic activity). The authors should carefully distinguish between “senolytic” and “senomorphic” terminology?

Response. We thank the reviewer for this thoughtful and important comment regarding the distinction between senolytic and senomorphic activities. We agree that some of our initial wording could overstate the evidence for “reversal of senescence” and have revised the manuscript to more carefully differentiate between the two activities.

Our results demonstrate that APPA exerts senotherapeutic activity against senescent human chondrocytes. Specifically, we observed a reduction of SA-β-Gal activity and p21 gene expression, together with an increase in cell numbers and the proliferation marker Ki67, which is consistent with a senomorphic effect. At the same time, APPA treatment also resulted in a decrease of SA-β-Gal–positive cells accompanied by a significant increase in apoptosis, particularly in cells with senescence induced by Eto-OSM, which points to a senolytic activity. Thus, APPA appears to combine both senomorphic and senolytic mechanisms.

In line with recent discussions in the field suggesting that the senolytic vs. senomorphic distinction may be “fuzzy” (PMID: 35781578), our findings support the idea that APPA should be best described as a senotherapeutic agent. This terminology reflects the dual nature of its effects: modulation of senescence-associated phenotypes and induction of senescent cell clearance. Specifically, our data show that T/C28a2 chondrocytes incubated with Eto+OSM and APPA displayed increased cell numbers, as well as elevated p-rps6 levels, suggesting recovery of proliferative capacity under a senescence-inducing environment. Furthermore, analysis of Ki67 expression revealed distinct cellular responses: in population P1, APPA further reduced proliferation when combined with Eto+OSM, while in population P2 APPA significantly increased the percentage of Ki67-positive cells (Figure 5C). These findings could be consistent with a senomorphic effect through attenuation of SA-β-Gal and partial restoration of proliferative potential. At the same time, we also observed a reduction in SA-β-Gal–positive cells and an increase in apoptosis in induced senescent cells, which could supports a senolytic effect. Taken together, these results indicate that APPA could exerts both senomorphic and senolytic activities, best captured under the broader definition of senotherapeutic activity.

We have revised the Discussion section to clarify this point including this new paragraph: “These data indicate that at least a fraction of chondrocytes exposed to Eto+OSM en-tered senescence and responded to APPA by enhancing apoptotic pathways, consistent with senolytic activity. In contrast, another subset of chondrocytes, likely in a different cellular state, responded by increasing Ki67 expression, suggesting a proliferative recovery more aligned with senomorphic effects. Taken together, our results suggest that APPA ex-erts senotherapeutic activity in senescent human chondrocytes through a combination of senomorphic and senolytic mechanisms. The reduction in SA-β-Gal activity together with increased cell numbers and Ki67 expression implies a senomorphic action, whereas the decrease in SA-β-Gal–positive cells accompanied by enhanced apoptosis, particularly under Eto-OSM–induced senescence, reflects a potential senolytic effect. These dual re-sponses align with recent perspectives that the boundary between senomorphic and senolytic activities is diffuse or “fuzzy” (76), and therefore support the classifica-tion of APPA as a potential senotherapeutic agent with dual activity.

  1. The manuscript briefly mentions osteoarthritis implications, but this needs to be developed. How could APPA be delivered in vivo? Would intra-articular injection or oral administration be feasible?

Response. Thank you for your question. We agree that the implications for OA warrant further clarification. APPA is currently under formal clinical development for OA and has already been tested in humans by oral administration. In a Phase 2a randomized, placebo-controlled trial in patients with symptomatic knee OA, oral APPA demonstrated both efficacy and safety (Bihlet et al., 2024; PMID: 38697511). Therefore, oral delivery is a feasible and clinically relevant route of administration.

Although intra-articular injection could theoretically be considered, this has not been explored in preclinical or clinical studies to date. Given the available data, oral administration remains the most validated and practical delivery method for APPA in vivo.

  1. Are there any prior reports of AP or PA in preclinical OA models that can be referenced?

Response. Thank you for your suggestion. There are indeed several preclinical studies testing the effects of AP or PA in both in vitro and in vivo models of osteoarthritis. To address your comment, we have added the following references: PMID: 16405885, PMID: 39659898, PMID: 2891096, and PMID: 28695367.

Additionally, we have revised the Introduction paragraph describing the characteristics of APPA and its components to reflect your suggestion: " Both AP and PA have shown efficacy in OA using in vitro and animal models (46-49). In the rat meniscus tear model of OA APPA has also been shown to be effective (50, 51) with evidence of synergy (unpublished). Randomized controlled trials in dogs with naturally occurring OA have provided evidence for the efficacy of APPA in this disease (52-54). In a recent Phase 2a study, oral administration of APPA demonstrated both efficacy and safe-ty, significantly reducing pain compared with placebo in a subset of patients with knee osteoarthritis”.

  1. 6. Pharmacokinetics, bioavailability, and potential toxicity in humans should be at least mentioned?

Response. Thank you for your suggestion. We did not focus on pharmacokinetics, bioavailability, or potential toxicity of APPA, as the present study was designed as an in vitro model rather than a clinical investigation. However, in the Phase 2a study by Bihlet et al., 2024 (PMID: 38697511), the authors reported on the efficacy and safety of APPA in symptomatic knee OA.

In the Introduction, we already referenced this study to highlight that APPA significantly reduced pain compared with placebo in a subset of patients: “In a recent Phase 2a study, oral administration of APPA demonstrated both efficacy and safety, significantly reducing pain compared with placebo in a subset of patients with knee osteoarthritis.”

To address your comment, we added the following sentence to further clarify safety: “In a recent Phase 2a study, oral administration of APPA demonstrated both efficacy and safety, significantly reducing pain compared with placebo in a subset of patients with knee osteoarthritis.”

  1. In the results section, avoid subjective language like “remarkably reduced” or “dramatically increased”; use quantitative descriptions?

Response. Thank you for your comment. We have revised the Results section to replace subjective language such as “remarkably reduced” or “dramatically increased” with precise, quantitative descriptions of the observed effects.

  1. The introduction could better integrate current literature on senotherapeutics in OA (e.g., studies with dasatinib + quercetin, fisetin, or navitoclax)?

Response. Thank you for your comment. Following your recommendation, we have incorporated two additional references (PMID: 38494091 and PMID: 39167845) and added a new paragraph in the Introduction to better integrate the current literature on senotherapeutics in OA:“Senotherapeutic agents under investigation for the treatment of OA include curcumin, dasatinib, quercetin, fisetin, navitoclax, and UBX0101. However, to date, none have demonstrated efficacy in clinical trials (29, 30).”

  1. Ensure consistency in abbreviations (e.g., APPA, SA-β-gal, ECM)?

Response. Thank you for your comment. We have carefully reviewed the manuscript and made efforts to ensure consistency in all abbreviations throughout the text, including APPA, SA-β-gal, and ECM. We hope that all abbreviations are now uniformly presented.

Reviewer 3 Report

Comments and Suggestions for Authors
  1. Parenthesis and abbreviation formatting is awkward. In “APPA (apocynin [AP] and pae- 2 onol [PA])”, the compound name for “pae- 2 onol” looks wrong (most likely “paeonol”). Also, the acronym “APPA” is not clearly explained. It’s implied to be a combination of AP and PA, but this isn’t made explicit.
  2. While the authors conclude that APPA exhibits both senomorphic and senolytic effects, the abstract provides limited mechanistic insight to clearly distinguish between these activities. Specifically, the simultaneous reduction in SA-β-gal activity and increase in apoptosis is suggestive but not definitive evidence of senolysis.
  3. In 4.1., the APPA formulation source and ratio are stated, but there is no justification for selecting the 2:7 AP:PA proportion.
  4. In 4.2, while patient demographics are provided, there is no mention of OA severity grading, which could affect cell characteristics and experimental variability.
  5. The DRAQ7 assay is used after 24 h treatment, but no justification is given for this time point, which may not capture delayed cytotoxic effects.
  6. SA-β-gal activity is quantified by flow cytometry, but no positive control for senescence.
  7. Only the Mann–Whitney U test is reported; it is unclear whether multiple comparison corrections were applied despite numerous group comparisons.
  8. The conclusion overinterprets SA-β-Gal reduction as evidence for SASP modulation and senomorphic activity without directly measuring SASP factors.

Author Response

Dear Reviewer-3

Thank you for your comments

  1. Parenthesis and abbreviation formatting is awkward. In “APPA (apocynin [AP] and pae- 2 onol [PA])”, the compound name for “pae- 2 onol” looks wrong (most likely “paeonol”). Also, the acronym “APPA” is not clearly explained. It’s implied to be a combination of AP and PA, but this isn’t made explicit.

Response. Thank you for your suggestion. In the text, we have referred to both the full names and chemical formulas of apocynin (4-hydroxy-3-methoxyacetophenone) (AP) and paeonol (2'-hydroxy-4'-methoxyacetophenone) (PA) at the end of the Introduction. In the title of the article, only the acronyms are used to keep it concise.

Several references describe APPA and its components as follows: Apocynin (AP) and paeonol (PA) are plant-derived phenolic compounds with notable anti-inflammatory and immunoregulatory effects, mainly through NF-κB, Nrf2, and related pathways. Their fixed combination, APPA (AP:PA = 2:7), has been shown to reduce reactive oxygen species, MMP-3, MMP-13, and senescent chondrocytes.

Following your recommendation, we carefully reviewed the manuscript for potential typographical errors in the spelling of paeonol, and these have now been corrected.

  1. While the authors conclude that APPA exhibits both senomorphic and senolytic effects, the abstract provides limited mechanistic insight to clearly distinguish between these activities. Specifically, the simultaneous reduction in SA-β-gal activity and increase in apoptosis is suggestive but not definitive evidence of senolysis.

Response. Thank you for your comment. To clarify our data in the abstract, we have modified the final paragraph to include a new conclusion: “This study suggests that APPA exerts senotherapeutic effects on human senescent chondrocytes. A reduction of SA-β-gal together with an increase in cell numbers and the proliferation marker Ki67 suggests possible senomorphic effects, whereas a reduction in SA-β-Gal accompanied by an increase in apoptosis indicates senolytic activity. These findings support recent evidence that the distinction between senolytic and senomorphic agents is ‘fuzzy’.”

  1. In 4.1., the APPA formulation source and ratio are stated, but there is no justification for selecting the 2:7 AP:PA proportion.

Response. Thank you for your comment. APPA, currently undergoing formal clinical development for the treatment of osteoarthritis, consists of the two phenolic compounds apocynin (AP) and paeonol (PA). The 2:7 AP:PA ratio was selected based on preclinical studies demonstrating optimal efficacy at this specific proportion. Animal studies showed that this formulation produced the most favorable therapeutic effects, which guided its selection for subsequent clinical testing (PMID: 38697511).

In our previous work (PMID: 38256951), APPA at the 2:7 ratio was shown to modulate multiple pathways implicated in the onset and progression of osteoarthritis. Moreover, studies investigating the effects of AP and PA on neutrophil function, including TNFα expression and signaling, further support the rationale for this combination (PMID: 3238062).

This fixed oral combination has also been evaluated in a phase 2a randomized, placebo-controlled trial for knee osteoarthritis, where APPA was administered at a dose delivering 88.9 mg of AP and 311.1 mg of PA—corresponding precisely to the 2:7 ratio (PMID: 38697511). The selection of this ratio was therefore informed by both preclinical efficacy and safety data, as well as tolerability observed in phase 1 studies.

Research indicates that the APPA combination modulates inflammatory pathways more effectively than either compound alone. Specifically, it inhibits NF-κB activation and enhances Nrf2 signaling, two central pathways in inflammation and oxidative stress regulation, thereby strengthening its anti-inflammatory efficacy in osteoarthritis models (PMID: 3238062, PMID: 38256951).

  1. In 4.2, while patient demographics are provided, there is no mention of OA severity grading, which could affect cell characteristics and experimental variability.

Response. Thank you for your question. You are correct that the manuscript did not specify the grade of OA, only that the donors underwent arthroplasty. To avoid variability across different stages of osteoarthritis and therefore in the results obtained, all donors included in this study were grade IV patients undergoing joint replacement. To clarify this, we have now specified the OA grade in the Methods section, modifying the sentence from “18 patients from total hip arthroplasty” to “18 patients from total hip arthroplasty (OA grade IV)”.

  1. The DRAQ7 assay is used after 24 h treatment, but no justification is given for this time point, which may not capture delayed cytotoxic effects.

Response. Thank you for your comment. The reason for using the DRAQ7 assay at a single 24-hour time point is that, in the experiments conducted in this manuscript, all treatments with AP, PA, and APPA were performed for 24 hours. Therefore, we only assessed cytotoxicity at this time point and did not investigate potential delayed cytotoxic effects. Evaluating longer-term cytotoxicity was beyond the scope of the present study, but we acknowledge that this could be addressed in future research

  1. SA-β-gal activity is quantified by flow cytometry, but no positive control for senescence.

Response. Thank you for your question. In the study, we used the incubation of chondrocytes with Eto + OSM as a positive control for senescence. When senescence was assessed under this condition, the results were positively modulated compared to the basal condition. It is true that in the manuscript, the comparison between the basal condition and the positive stimulus (Eto + OSM) is only shown in Figure 1, panels A and C. This was because we wanted to focus on the effects of the evaluated compounds (AP, PA, and APPA) on senescent chondrocytes, once the ability of Eto + OSM to increase SA-β-gal activity in chondrocytes had been established.

  1. Only the Mann–Whitney U test is reported; it is unclear whether multiple comparison corrections were applied despite numerous group comparisons.

Response. Thank you for your comment. As described in the manuscript, group comparisons were performed using the non-parametric unpaired Mann–Whitney U test. Comparisons were made specifically versus the basal (untreated) condition or versus the Eto + OSM condition, as indicated by the symbols in the figures. No additional corrections for multiple comparisons were applied, as each comparison was planned a priori and conducted independently. We acknowledge that applying multiple comparison corrections could further control for type I error, and this will be considered in future studies.

  1. The conclusion overinterprets SA-β-Gal reduction as evidence for SASP modulation and senomorphic activity without directly measuring SASP factors.

Response. Thank you for your comment. We agree that SA-β-Gal is not a SASP factor however in our previous publication (PMID: 38256951) we have shown that APPA reduced the gene expression of IL-8, TNF-α, MMP-13 and MMP-3 all SASP markers. To avoid any misunderstanding, we have revised the corresponding part of the conclusion. It now reads: “Reduction of SA-β-gal activity, together with an increase in cell numbers and the proliferation marker Ki67, suggests…”